# Position: The Open Benchmark Paradox
# Must be Resolved through Sovereign Medical Evaluation

**Keonwoo Kim** [1,2]  **Hyeseon Ko** [1]  **Hyejeong Jo** [1]  **Sewon Kim** [1]  **Yera Choi** [1]  **JaeDeok Lee** [1]  **Heeyoung Kwak** [1]
**Yunwook Sung** [3]  **Haanju Yoo** [1]

## Abstract

As generative medical large language models become increasingly involved in clinical actions, public benchmarks are often treated as proxies of deployment-readiness. However, this reliance creates a false sense of security because public scores are often based on data the models have already seen. We call this the **Open Benchmark Paradox**: making evaluation data public for research progress also makes data contamination inevitable, ruining its value as a reliable safety signal. This paradox induces three structural failures: (1) *hidden contamination*, where it is impossible to prove evaluation independence; (2) *outdated standards*, where static datasets fail to track evolving medical guidelines; and (3) *jurisdictional divergence*, where global averaging ignores local legal and ethical standards. To validate these risks, we audited frontier models using recent medical exam data, which confirmed a high probability of data contamination. To resolve such integrity issues in medical evaluation, we propose Sovereign Medical Evaluation (SME). Instead of public leaderboards, SME establishes a national infrastructure where health authorities manage private, isolated evaluation pipelines. Within this secure system, evaluations are automatically updated using live medical data and legal changes, ensuring they remain current and strictly separated from model training. SME provides the essential transition to a controlled, auditable, and legally grounded safety gate for clinically deployed generative medical LLMs.

[1]NAVER Applied AI Group [2]Seoul National University [3]Healthcare AI Researcher, Seoul National University Hospital. Correspondence to: Haanju Yoo <haanju.yoo@navercorp.com>.

*Proceedings of the 43rd International Conference on Machine Learning*, Seoul, South Korea. PMLR 306, 2026. Copyright 2026 by the author(s).

## 1. Introduction

Performance on medical benchmarks is frequently treated as a proxy for clinical competence in medical AI research (Nori et al., 2023; Singhal et al., 2023). Following the precedent set by Med-PaLM and subsequent systems (Singhal et al., 2023; Tu et al., 2024; Singhal et al., 2025), high scores on licensing-style exams are increasingly seen by some as an indicator of readiness for clinical use. While these evaluations are useful for measuring medical knowledge, they are now often used as informal measures of clinical safety, a shift we contend lacks a reliable basis.

This trend appears to rest on a premise that may not fully account for the complexities of clinical application. The ability to answer multiple-choice questions or solve curated clinical vignettes is not the same as safe clinical practice (Chen et al., 2024b; Singhal et al., 2025). Actual practice requires more than factual accuracy; it demands strict adherence to the procedural, legal, and ethical standards that govern real-world care (Organization, 2024). We argue that this gap between test success and deployable safety makes current benchmarks an unreliable measure of a model's readiness for the clinic.

Recent industry benchmarks (Fleming et al., 2024; Wang et al., 2025; Nori et al., 2025; Arora et al., 2025) attempt to move beyond simple exam-style questions by introducing realistic clinical scenarios. While these efforts improve evaluation content, they still face a structural problem we define as the **Open Benchmark Paradox**: the public availability of benchmark data makes it nearly impossible to ensure that a model's performance is independently verified (Raji et al., 2020).

In safety-critical domains where patient safety is paramount, a valid evaluation must be independently auditable, dynamically updated, and jurisdictionally aligned. In the absence of such a framework, clinicians risk prioritizing models based on inflated, contaminated scores over actual clinical reliability. Current public benchmarks (Jin et al., 2019; 2021; Pal et al., 2022; Kweon et al., 2024; Pal et al., 2024) fail to meet these requirements. We argue that the Open Benchmark Paradox leads to three core failures.

First, public benchmarks suffer from a *hidden contamination*. Because frontier models are trained on massive, non-transparent datasets, it is impossible to verify whether the model encountered the test questions during its training (Zhou et al., 2023; Sainz et al., 2023; Dong et al., 2024; Xu et al., 2024; Deng et al., 2024). High scores may reflect memorization rather than genuine clinical skill, leading to misleading results. Second, *outdated standards* occur because static public datasets cannot keep pace with evolving medical guidelines. Any public update to a benchmark immediately exposes the new data to the risk of being ingested by training pipelines (Lazaridou et al., 2021). This creates a cycle where updating a public dataset leads to further data contamination. Third, openness leads to *jurisdictional divergence* (Lee, 2024; Mello & Cohen, 2025). While regional datasets exist to address local needs (Jin et al., 2021; Pal et al., 2022; Kweon et al., 2024), the pressure to maintain high global leaderboard rankings incentivizes model providers to emphasize a global average over specific local requirements. In this race for scores, the nuanced reasoning needed for local laws or cultural ethics is often replaced by memorization of the most common global patterns. This creates an averaging trap, where models favor public data over genuine adherence to local standards of care.

These structural limitations have practical consequences, as clinicians increasingly use LLMs for learning and decision support (Blease et al., 2024). In the absence of jurisdictional oversight, unverified or contextually inappropriate information may be integrated into clinical practice, potentially affecting the quality and safety of patient care. This growing reliance on uncertified systems suggests that the lack of a reliable evaluative framework is no longer just a technical oversight, but a critical vulnerability in clinical safety.

**Position: We argue that public medical benchmarks are structurally inadequate for certifying the clinical safety of generative medical LLMs intended for deployment. Evaluation of such systems must therefore shift from global leaderboards toward *Sovereign Medical Evaluation*: a national safety infrastructure that is jurisdictionally governed, auditable, and continuously updated.**

## 2. The Categorical Uniqueness of Medical AI Evaluation

To understand why public benchmarks fail, we must first define what medical evaluation is intended to measure. In fields like mathematics or coding, correctness is often absolute and binary (Ahn et al., 2024; Jiang et al., 2024; Guo et al., 2025). In medicine, however, safety extends beyond mere factual accuracy; it requires clinical judgment to prevent harm. Therefore, medical AI evaluation must shift from testing general medical knowledge to verifying safe behavior that follows local regulations and accountability.

### 2.1. Knowledge vs. Clinical Action

Some researchers view success on medical licensing exams as an indicator of clinical competence. However, in medicine, knowing a fact is not the same as taking an action (Thapa et al., 2025). Even when medical models (Chen et al., 2024a; Wu et al., 2025) exhibit high recall of facts, they often fail to maintain procedural adherence, particularly in tasks requiring the application of complex clinical protocols. A model might explain a disease correctly while failing to follow local triage rules or legal duties. Evaluating a medical AI as if it were a general knowledge tool creates a gap between test scores and actual safety. Evidence from interactive simulations (Schmidgall et al., 2024) further highlights the discrepancy, showing that models with superior scores on static benchmarks often struggle to maintain clinical agency and diagnostic accuracy.

### 2.2. Structural and Linguistic Divergence

Medical terms are not universal, unlike mathematical symbols, as they often carry distinct meanings across different healthcare systems (Cimino, 1998; Rector, 1999). Even standardized frameworks, such as the International Classification of Diseases (ICD) (WHO, 1992) or the Fast Healthcare Interoperability Resources (FHIR) (Bender & Sartipi, 2013), rely on national modifications and localized extensions to be functional. This structural divergence results in different applications of the same diagnostic terms across countries (Jetté et al., 2010; Otero Varela et al., 2021), meaning that model proficiency in one jurisdiction does not ensure structural compatibility under standardized benchmarks.

To ensure repeatability, public benchmarks often homogenize evaluation data by removing localized nuances, including shorthand (Arora et al., 2025), administrative acronyms, and regional jargon common in real-world settings. By decontextualizing the data, benchmarks fail to assess a model's ability to navigate communication protocols and clinical pathways within a given jurisdiction. Consequently, these evaluations measure performance on abstract language tasks rather than the procedural, legal, and urgent requirements of real healthcare systems. This oversight creates a false sense of security, as a model may excel at textbook medicine while remaining ill-equipped for localized clinical practice.

### 2.3. Jurisdictional Alignment as a Hard Constraint

Clinical practice is jurisdiction-dependent. Although disease biology is universal, diagnostic and therapeutic standards are defined by national guidelines, available drugs, and local norms. As a result, identical pathophysiology can lead to different safe treatments across countries. For instance, the management of Chronic Myeloid Leukemia (CML) exemplifies this divergence (Apperley et al., 2025). International

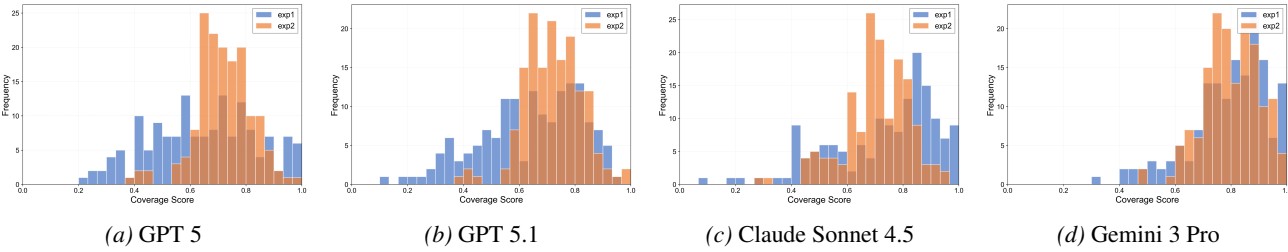

| *(a)* GPT 5 | *(b)* GPT 5.1 | *(c)* Claude Sonnet 4.5 | *(d)* Gemini 3 Pro |

*Figure 1.* Distribution of coverage scores across frontier models. Blue and orange denote results from Experiments 1 and 2, respectively.

frameworks such as the European LeukemiaNet recognize six tyrosine kinase inhibitors (TKIs), imatinib, dasatinib, bosutinib, nilotinib, ponatinib, and asciminib, as standard first-line options (Kantarjian et al., 2025). In South Korea, however, the Ministry of Food and Drug Safety (MFDS) has approved radotinib, a second-generation TKI, for newly diagnosed or refractory chronic-phase CML (Kwak et al., 2017). This agent remains unlicensed in most Western countries, illustrating how local approvals can differ markedly from global recommendations. Such disparities mean that AI models trained on Western clinical data might mistakenly flag locally sanctioned treatments like radotinib as non-standard. Effective decision support, therefore, requires explicit alignment with jurisdiction-specific guidelines to avoid inappropriate or unsafe recommendations.

This diversity dictates that clinical safety cannot be measured by a single global metric. A model may achieve expert-level performance on a shared global leaderboard while simultaneously violating local statutory protocols (Mwaniki et al., 2025; Ke et al., 2025). Standard evaluations often combine different statutory and ethical rules into a single score. We call this the averaging trap, where a high global score hides a failure to follow the standards of a specific jurisdiction. Since clinical correctness is location-dependent, localized alignment remains an absolute safety requirement. Evaluations bypassing jurisdictional differences are structurally deficient, as they ignore the regulatory standards that define the boundaries of safe care.

## 3. The Open Benchmark Paradox

Public benchmarks facilitate research progress through open comparison, but they are not intended to be safety systems. When these benchmarks are used to decide if a model is ready for actual use, a major problem appears. The openness that helps research makes it impossible to find and fix safety errors. This section explains three main failures caused by this paradox that cannot be fixed in an open system.

### 3.1. Hidden Contamination

While benchmark contamination is a widely recognized technical challenge (Zhou et al., 2023; Golchin & Surdeanu,

2023; Sainz et al., 2023; Dong et al., 2024; Xu et al., 2024; Deng et al., 2024), the deeper issue is that we can no longer prove if an evaluation is truly independent. Once a benchmark is public, it likely becomes part of the massive web data used to train models. In this open system, high scores might reflect past exposure rather than genuine skill. We call this *hidden contamination*. To investigate this without internal access to the models, we used the 2025 Korean Medical Licensing Examination (KMLE) as our primary audit target. We chose the 2025 KMLE for two reasons:

(1) *Signal Isolation*: Most existing benchmarks rely on widely used languages such as English or Chinese that are prevalent in training corpora. Evaluating models on Korean, a comparatively less frequently used language, is expected to reduce the influence of prior training exposure. (2) *Benchmark Independence*: Public and curated benchmarks such as KorMedMCQA (Kweon et al., 2024) may be susceptible to training data contamination, a concern often associated with the Open Benchmark Paradox. In contrast, we manually collected raw questions from the 2025 KMLE (Korea Health Personnel Licensing Examination Institute, 2025), which had not been processed into benchmark-style datasets, and thus are less likely to be contaminated.

To check for training data exposure, we use N-gram coverage analysis (Hallinan et al., 2025), a memorization-based probe related to membership inference for API-based black-box models. The core intuition is that models reproduce seen text with substantially higher fidelity than unseen text. We conducted two experiments to test this (detailed settings are in Appendix A).

- **Experiment 1 (Choice Reconstruction):** provide the question as input and ask the model to generate the original multiple choice options. We then measure how much the generated text overlaps with the ground truth.

- **Experiment 2 (Text Completion):** We provide only the first 50% of the question text and ask the model to complete the remaining 50%.

The results in Table 1 provide compelling evidence of data exposure across GPT-5, GPT-5.1, Claude Sonnet 4.5, and

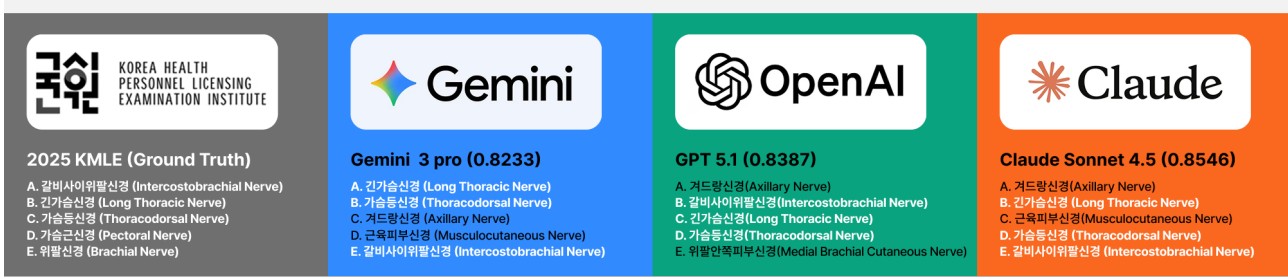

*Figure 2.* Qualitative results from Experiment 1. Ground Truth corresponds to the original choices from the official exam; values in parentheses next to each model name indicate the coverage score, and white-highlighted options denote overlap with the Ground Truth.

*Table 1.* N-gram coverage analysis on the 2025 KMLE. Acc. denotes downstream task accuracy.

| Model | Exp 1 (Percentile) | | Exp 2 (Percentile) | | Acc. (%) |
|---|---|---|---|---|---|
| | 95th | 99th | 95th | 99th | |
| GPT 5 | 0.9518 | 1.0000 | 0.8688 | 0.9217 | 93.62 |
| GPT 5.1 | 0.8854 | 0.9295 | 0.8699 | 0.9708 | 95.99 |
| Claude Sonnet 4.5 | 0.9830 | 1.0000 | 0.8646 | 0.9200 | 94.86 |
| Gemini 3 Pro | 1.0000 | 1.0000 | 0.9489 | 0.9966 | 98.23 |

Gemini 3 Pro (OpenAI, 2025; Anthropic, 2025a; G. Deep-Mind, 2025b). In Experiment 1, multiple models achieve a perfect coverage score of 1.0000 even at the 95th percentile, while Experiment 2 similarly yields exceptionally high values, suggesting that the models possess near verbatim knowledge of the test items. Beyond these top percentile metrics, the coverage distributions in Figure 1 reveal the systemic scale of data exposure; the scores for frontier models exhibit a pronounced left tail pattern. This distribution confirms that clustering at the higher end of the spectrum is a dominant characteristic across the entire dataset rather than an isolated occurrence.

The qualitative evidence in Figure 2 further reinforces these quantitative findings. Even in cases where the coverage score is approximately 0.8, all models successfully reconstruct at least three out of five original multiple-choice options. We interpret this observation as potentially indicative of a rapid pace of automated training data collection.

While these results provide strong empirical evidence, we acknowledge that membership inference methods in a black box setting cannot offer absolute certainty of contamination. However, we argue that this inherent non-verifiability is the fatal flaw of the open benchmark model (Cheng et al., 2025). In high-stakes domains like medicine, the mere pos-

sibility of hidden contamination creates an unacceptable uncertainty that public leaderboards are structurally unable to resolve. Since the independence of an open evaluation can never be definitively proven or protected against automated collection (Jacovi et al., 2023), the open benchmark paradigm is incapable of preserving its evaluative integrity. Once test data is released to the public web, the resulting loss of independence becomes permanent, rendering static scores unreliable as continuous safety signals.

### 3.2. Outdated Standards

Medical knowledge evolves continuously with new clinical evidence (Shekelle et al., 2001). While some benchmarks specify validity periods (García et al., 2014), most remain static snapshots. This creates a temporal gap where model performance reflects outdated standards rather than current clinical reality. For example, a treatment recommended by a 2021 benchmark might be considered obsolete or even unsafe by 2026 standards. Consequently, static public benchmarks act as lagging safety signals, failing to provide the continuous assurance needed in a field where medical consensus is constantly updated.

### 3.3. Jurisdictional Divergence and the Averaging Trap

Clinical safety depends on local laws and rules, but public benchmarks often combine these different standards into a single score. Even sophisticated efforts like Health-Bench (Arora et al., 2025), which aim to provide a unified framework for medical performance and safety, encounter limitations in diverse clinical settings, including mismatches with local practices, as evidenced by suicide-risk scenarios that reference non-local emergency services in the Japanese context (Hisada et al., 2025). Recent analysis (Hisada et al.,

2025) also has shown that such global benchmarks may fail to reflect the specific clinical and regulatory realities of localized healthcare systems, as seen in the discrepancies between global safety metrics and Japan's medical landscape.

We argue that openness transforms local alignment into a non-verifiable recitation task. Since jurisdictional constraints (*e.g.,* local medical laws) are publicly accessible, models can memorize evaluation artifacts rather than demonstrate genuine adherence in unseen scenarios. This averaging trap allows models to pass global leaderboards while carrying the inherent risk of violating local triage protocols or legal mandates. It reduces jurisdiction-specific alignment to a mere nominal compliance, as a high global score can mask critical failures in following local safety standards.

### 3.4. Design Requirements

The limitations of public benchmarks define four structural requirements for medical evaluation framework: **(R1) Data Isolation** ensures strict separation from training pipelines and automated crawlers; **(R2) Controlled Governance** provides auditable access managed exclusively by authorized regulators; **(R3) Temporal Currency** maintains alignment with evolving medical knowledge and clinical guidelines; and **(R4) Jurisdictional Alignment** mandates compliance with regional legal and ethical frameworks.

## 4. Position: Sovereign Medical Evaluation

We propose **Sovereign Medical Evaluation (SME)** as a national safety infrastructure, which we see as a necessary step to resolve the Open Benchmark Paradox. In this context, we define national as the minimum governance unit capable of enforcing legal accountability and clinical standards, rather than a requirement for centralized state control. Individual healthcare institutions are important stakeholders, yet they lack the jurisdictional authority to issue legally binding safety certifications or the scale needed to maintain reliable, up-to-date evaluation systems. SME, therefore, anchors assessment within the sovereign framework that governs medical liability and the standard of care. Our central argument is that clinical safety signals are no longer reliable within an open ecosystem (Section 3). To restore evaluative validity, we must transition to a framework that physically and institutionally separates assessment from the web-scale training pipelines used by model developers.

**Scope and Risk-Tiered Applicability.** We explicitly narrow the focus of SME to *generative medical LLMs intended for clinical deployment*, rather than to AI-enabled medical devices such as radiology classifiers, whose narrowly scoped outputs already fall under established device-validation frameworks such as the FDA guidelines for AI/ML-enabled medical devices. Generative medical LLMs, by contrast,

implicitly propose clinical reasoning and treatment plans whose outputs are not locally verifiable, reflecting a shift from evidence provider to reasoning agent that renders static validation frameworks insufficient. SME is not intended as a universal requirement: high-stakes clinical deployment (e.g., autonomous triage, dosing suggestions, diagnostic assistants) requires SME, while low-stakes research and education (e.g., literature summarization, information extraction, non-deployment prototypes) remains under existing institutional review with open benchmarks as developmental signals.

### 4.1. Controlled Infrastructure

To address the problem of hidden contamination, where evaluation independence cannot be easily verified (Section 3.1), SME establishes a controlled infrastructure for model assessment (Table 2). This approach ensures that independence is a governed, verifiable state, thereby satisfying **R1** and **R2**.

**Isolation through Evaluation-only Access.** SME replaces public benchmark release with an evaluation-only access model. Through physical and legal isolation, model providers are denied access to raw test samples; instead, they submit inference interfaces to a protected environment managed by a jurisdictional authority. To mitigate membership inference attacks, the system restricts request quotas and feedback frequency, preventing the repetitive probing used to extract latent data. By breaking the Open Benchmark Paradox loop, SME ensures evaluation remains a strictly non-training channel (**R1**). It mirrors pharmaceutical safety testing (Carpenter, 2014), where private criteria prevent gaming and ensure the reliability of the safety signal.

**Mid-party Operationalization.** A practical concern is that model providers regard their weights and serving stacks as proprietary assets, and may be unwilling to deliver them directly to a state evaluator. SME accommodates this constraint through a *mid-party* arrangement, in which evaluation is brokered by an accredited intermediary rather than executed by the state itself, facilitating governance over the evaluation *process* rather than requiring state ownership of compute or model weights. In practice, this can be operationalized through accredited third-party evaluators, or through provider-hosted confidential endpoints that operate under jurisdictional control. Either mode resolves the tension between protecting model intellectual property and ensuring robust jurisdictional oversight, allowing SME to govern the conditions of evaluation without taking custody of providers' inference infrastructure.

**Operational Protocol.** To make the mid-party arrangement auditable in practice, SME defines an operational protocol comprising three elements:

*Table 2.* Comparison between Public Medical Benchmarks and Sovereign Medical Evaluation (SME). The table illustrates how SME addresses the structural failures of the Open Benchmark Paradox by satisfying four core requirements (R1–R4).

| Requirement | Key Feature | Public Benchmarks | Sovereign Medical Evaluation (SME) |
|---|---|---|---|
| **R1. Data Isolation** | Data Privacy | ✖ (Open Access) | ✔ (Isolated API Access) |
| **R2. Governance** | Auditability | ✖ (Unverifiable) | ✔ (Verifiable Audit Trail) |
| **R3. Temporal Currency** | Update Frequency | ✖ (Static Snapshots) | ✔ (Dynamic Live Updates) |
| **R4. Jurisdictional** | Alignment Target | ✖ (Global Averaging) | ✔ (Local Legal Mandates) |

- **Pre-registered evaluation manifest.** Providers formally declare model parameters (e.g., version hash, tokenizer, and hardware tier) before evaluation begins.

- **Dry-run validation.** Providers conduct preliminary execution on non-secret calibration tasks to verify pipeline compatibility with the SME environment.

- **Attested execution logs.** While test items remain private, all execution states and resource usage are logged in an attested form to support auditability.

The computational burden therefore remains with the provider, so providers can run frontier-scale models on their own optimized infrastructure, mitigating capacity constraints on the evaluator's side while keeping evaluation conditions transparent and legally auditable.

**Verifiable Governance and Auditability.** Unlike static leaderboard scores, SME defines safety assessment as an inherently auditable process. Since SME centralizes the inference environment by routing model queries through a protected, regulator-managed gateway, the system inherently generates a verifiable audit trail of evaluation interactions (**R2**). This structural linkage transforms evaluation from a one-time result into a forensic record. It enables regulators to investigate not just the final output, but the specific reasoning patterns, such as the verbatim reconstruction documented in Figure 2. By restoring the verifiability of independence, SME rebuilds evaluative integrity and eliminates contamination introduced through the evaluation channel itself, providing a rigorous demonstration of clinical performance.

### 4.2. Continuous Evaluation and Temporal Currency

To address the issue of outdated standards, where static datasets fail to keep pace with evolving medical knowledge (Section 3.2), we propose a dynamic evaluation system. This approach replaces frozen test sets with an automated pipeline, which is overseen by clinical experts, that derives evaluation scenarios from live, authoritative national sources (**R3**) (Granlund et al., 2024). Unlike public updates that immediately leak into training pipelines, this dynamic framework manages the transition from medical knowledge to clinical reasoning within the protected environment of the SME. The pipeline ensures that safety assessment remains current through a three-stage process:

**Source Acquisition and Triggering.** The system monitors jurisdictional sources, which are managed directly by the regulatory bodies that issue clinical mandates, such as national food and drug safety agencies (e.g., MFDS) or centers for disease control. Any change in these authoritative sources triggers an immediate refresh of the evaluation, such as a new black-box warning (Lasser et al., 2002) for a specific medication. State operation is necessary because only a jurisdictional authority has the legal mandate to translate regulatory updates into binding safety constraints. Delegating this responsibility to private or fragmented institutions would weaken the chain of clinical accountability. This structure ensures that the pace of evaluation matches the evolution of medical standards.

**Expert-led Case Development.** The system converts regulatory updates into clinical scenarios designed to test safety-critical reasoning. Instead of testing for factual recall, such as identifying a new drug contraindication, the evaluation assesses whether a model follows updated rules in complex clinical situations. For instance, the system evaluates whether a model appropriately withholds medication when presented with emergent risk factors. By generating unique patient profiles and counterfactual cases absent from public domains, the SME demands genuine reasoning rather than simple memorization. This human-in-the-loop approach, validated by jurisdictional experts, ensures that evaluations remain clinically rigorous and focused on reasoning instead of historical pattern matching.

**Temporal Versioning.** Each evaluation report is timestamped and linked to specific versions of the source guidelines. This systematic versioning provides the necessary foundation for accurate performance comparisons between different models or subsequent iterations of the same system. Because these evaluation scenarios remain within the protected environment and are never released publicly, the framework effectively eliminates the risk of hidden contamination. The results serve as a rigorous demonstration of a

model's actual clinical performance, ensuring that improvements reflect genuine enhancements in capability rather than the mere acquisition of updated training data.

**Three-tier Stratified Rolling Protocol (orthogonal content axis).** Whereas the three stages above describe *how* scenarios are produced, the protocol below describes *what content* the evaluation suite must cover. The content produced by Expert-led Case Development is organized along three complementary strata, reflecting the heavy-tailed prevalence distribution of clinical cases:

- **Tier 1 (Temporal stratum).** For newly approved drugs and updated clinical guidelines, a strict time-cliff protocol is enforced so that models are evaluated on information that post-dates their training sets.

- **Tier 2 (Core stratum).** For high-prevalence diseases, the rolling property is maintained by procedurally varying clinical vignettes (e.g., patient demographics, comorbidities, presenting symptoms) without altering the underlying diagnosis, thereby preventing reliance on memorized templates.

- **Tier 3 (Adversarial stratum).** Atypical presentations and rare contraindications are continuously incorporated to evaluate safety limits rather than pattern recall.

By integrating these mechanisms, SME establishes a sustained evaluation framework via rolling datasets, creating a continuous mitigation cycle that systematically bounds the risk of data contamination and memorization.

### 4.3. Jurisdictional Governance

Sovereign Medical Evaluation (SME) defines jurisdictional alignment as a governed property that resolves the global averaging trap (Section 3.3), rather than a passive metric. By anchoring evaluation within a national legal framework, SME shifts safety assessment from an informal task of checking for data recitation into a mechanism of regulatory accountability (**R4**).

**Encoding Regional Mandates.** SME integrates jurisdictional mandates, such as local triage protocols, drug availability, and specific medical liability laws, directly into its evaluation criteria. This is in contrast to public benchmarks, which often aggregate various standards into a single score. This approach ensures that alignment serves as a hard constraint. A model must demonstrate that it follows the specific standard of care for the jurisdiction where it is used, rather than relying on a globally averaged knowledge base that might violate local legal or ethical requirements.

**Institutional Integrity through Governance.** The reliability of the safety signal depends on a governance-led architecture where jurisdictional alignment is verifiable. Within the SME, the audit trail is not merely an optional log but a foundational output of the system's role-based access control. Every interaction is timestamped and linked to specific versions of clinical mandates, allowing authorities to confirm whether model responses derive from genuine reasoning or the memorization of leaked samples. To prevent internal compromise, SME mandates a functional separation of duties between source acquisition, case development, and execution teams. By limiting the visibility of any single party in the complete evaluation suite, the system ensures that the assessment results cannot be gamed.

**Regulatory Authority and Liability.** The primary output of SME is a safety certification with legal weight, rather than a simple leaderboard rank. By providing a verifiable and continuously updated measure of jurisdictional alignment, SME allows oversight bodies to issue deployment permits and establish clear liability frameworks. Establishing this formal baseline for clinical safety creates a reference for liability allocation, which clarifies the scope of institutional responsibility if a clinical failure occurs. This transitions medical AI evaluation from a research-oriented comparison to a functional safety gate, ensuring that clinical agents are both technically capable and legally compliant with the specific systems they serve.

## 5. The Sovereign Ecosystem

SME functions as an infrastructure that connects safety assessment with the regulatory frameworks governing clinical practice. In the medical domain, evaluation signals are most effective when they are integrated into systems for accountability and monitoring. This section examines how a sovereign infrastructure supports safe real-world operation by linking assessment to clinical governance.

### 5.1. Auditability and the Liability Regime

Clinical deployment requires a clear chain of accountability. When AI-assisted decisions result in harm, the legal system must determine the assignment of the duty of care. Although establishing verifiable audit trails through SME involves technical overhead and requires social consensus, such mechanisms are necessary because public benchmarks lack the transparency needed for legal responsibility.

SME addresses this gap by providing law-governed audit records that establish a verifiable standard of care for AI interventions. These records enable investigators to distinguish between model failure, data drift, or user error, allowing regulators to set enforceable deployment conditions. By providing the evidentiary grounding for a functional liability regime, SME transforms medical AI from an opaque black box into a governed component of national infrastructure.

## 5.2. Continuous Safety: Monitoring the Model Lifecycle

Point-in-time assessments are insufficient for medical AI, vulnerable to data distribution shifts and evolving clinical standards. SME facilitates a continuous safety lifecycle by linking dynamic evaluation to live jurisdictional triggers, such as new drug safety alerts, transitioning evaluation from a static snapshot into an active monitoring process (**R3**). By integrating incident reports and near-miss data from clinical sites, the framework can establish a responsive feedback loop that adapts to real-world failure patterns. This ensures that the safety profile of deployed models remains synchronized with national infrastructure and current clinical reality.

## 5.3. Regulatory Co-design and Market Alignment

Medical AI regulation often struggles with the gap between research-level claims and clinical safety requirements. SME facilitates regulatory co-design, aligning evaluation protocols directly with clinical guidelines and procurement standards. This alignment ensures that jurisdictional compliance becomes a prerequisite for market entry. By mandating SME reports for adoption, health systems force a shift in provider priorities from public benchmarks to local safety and real-world reliability. Removing the guarantee of clinical adoption based on global scores naturally redirects AI development toward practical clinical utility.

## 5.4. Technical Sovereignty

As medical AI shapes population health and handles sensitive data, it functions as critical national infrastructure. Relying on external, non-transparent evaluation signals creates a dependency where a nation cannot independently verify the reliability of its essential healthcare services. Sovereign evaluation mitigates this risk by establishing secure execution environments and rigorous audit procedures. This framework formalizes the role of AI within national safety systems, enabling effective incident response and reducing reliance on proprietary metrics that lack transparency during systemic failures. By treating evaluation as infrastructure, the state can maintain sovereign oversight over the safety and reliability of the healthcare services.

## 6. Call to Action: A Strategic Roadmap

Establishing SME requires coordinated action between developers, assessors, and regulators. To shift from leaderboard competition to jurisdictional safety assurance, we propose a roadmap focused on building essential institutional and technical infrastructure.

## 6.1. Regulators and National Health Authorities

For high-stakes clinical deployments, the state must provide the foundation for SME through a structured approach. First, regulators should mandate evaluation-only API access within secure, jurisdictionally-governed perimeters (per Section 4.1) to ensure strict data isolation (**R1**). Second, evaluation scenarios and artifacts must remain confidential, with penalties for unauthorized ingestion into training data to prevent contamination and ensure the independence of the process (**R2**). Third, these assessments should be conducted through dynamic evaluation cycles rather than static tests to ensure they remain current knowledge (**R3**). Finally, national health authorities should be designated to issue formal safety certifications, as only state-authorized bodies have the jurisdictional mandate to transform technical results into the legal standards that define the standard of care (**R4**).

## 6.2. Health Systems and Professional Societies

Hospitals and professional organizations reinforce this framework by aligning their procurement and documentation processes. Hospital consortia should mandate jurisdictional safety audit reports as a prerequisite for adoption, shifting the evaluative focus from global marketing claims to local safety performance. Furthermore, professional societies should prioritize machine-readable documentation to facilitate rapid, expert-led updates to evaluation scenarios. This transition to structured clinical data ensures the system remains synchronized with the latest medical standards.

## 6.3. Academic Community as Primary Architects

A common concern is that national-level infrastructure may marginalize the academic community. We argue the opposite: SME is most effective when academics serve as its primary scientific architects rather than as administrative operators. While state authorities provide the jurisdictional mandate and secure hosting, the academic community is responsible for designing the scientific core of SME, including automated red-teaming pipelines, dynamic scenario-generation methods, and contamination-detection heuristics. These contributions are integrated under the role-segregated access regime described above, preserving the separation of duties between source acquisition, case development, and execution. Importantly, SME does not displace open science: it sits within a multi-stage ecosystem in which open benchmarks remain the primary venue for foundational academic R&D, while SME provides the specialized regulatory gateway through which clinical certification is established. This division of labor preserves academic inquiry and early-stage innovation while maintaining a rigorous standard for clinical deployment.

## 6.4. Model Providers and Industry

Industry leaders should adopt architectures that respect jurisdictional boundaries and data isolation. To comply with isolation requirements, providers should offer model endpoints within managed jurisdictional clouds that are technically prevented from learning from evaluation queries (**R1**). Additionally, the industry should provide verifiable reasoning logs for each assessment. These logs allow authorities to examine the clinical reasoning path of a model and investigate specific failure modes. By utilizing verification methods such as MIA, regulators can further ensure that model responses are based on genuine reasoning rather than the mere reconstruction of memorized training data (**R2**).

# 7. Alternative Views

An alternative perspective suggests that global standardization remains the most efficient and transparent pathway for medical AI development. Critics of the sovereign approach argue that jurisdictional focus may lead to fragmentation, which could hinder scientific progress, exacerbate global health inequities, and foster institutional opacity.

## 7.1. Standardization and Scientific Progress

This perspective argues that core medical knowledge is inherently universal. Since fundamental clinical principles and medical facts remain consistent across borders, standardized benchmarks like MedQA (Jin et al., 2021) provide a common language for the research community. This allows foundation models to scale globally and facilitates shared scientific progress. From this perspective, Sovereign Medical Evaluation (SME) might create fragmented silos that hinder the collective growth driven by global leaderboards.

**This perspective, however, overlooks the distinction between factual accuracy and clinical safety.** A model may demonstrate medical knowledge while failing to follow specific local protocols or legal requirements. While research benchmarks are useful for measuring technical progress, they cannot certify safety. Safety is not a universal metric, but a performance grounded in local clinical and legal standards. As elaborated in Section 6.3, SME complements rather than displaces open science for safety certification; it defines where scientific validity must align with jurisdictional responsibility. Ultimately, standardization allows for comparison, but safety requires local oversight.

## 7.2. Resource Constraints and Equity

Another critique focuses on the resource requirements of sovereign infrastructure. Establishing such a framework requires clinical expertise, curated data, and secure compute environments. There is a legitimate concern that emphasiz-

ing jurisdictional sovereignty could create a gap where only high-income nations can certify AI systems, potentially excluding low-resource regions from the benefits of frontier models or forcing them to rely on uncertified systems.

**While these concerns are valid, they assume that jurisdictional evaluation requires entirely independent development.** Instead, safety can be achieved through collaborative models where international organizations provide open-source technical frameworks for SME. Individual nations can then apply their specific clinical and legal rules to these shared systems. This approach allows low-resource regions to utilize evaluation frameworks without the high costs of independent research and development, ensuring that equity is achieved through shared technology rather than lower safety standards. SME does not reduce transparency; it adapts it to an environment where process-level verification is necessary to ensure reliable safety.

## 7.3. Audit Trails and Participation Incentives

A major concern regarding the audit trail is the incentive for model providers to participate. Critics may argue that private firms will avoid documenting potential failures to prevent such records from being used in future litigation. Companies might prefer the ambiguity of current benchmarking practices to avoid direct accountability for specific model errors. **However, we believe this framework actually offers a clear path to managing legal risk.** Current public benchmarks provide no evidence that a company exercised due diligence if a model fails in a clinical setting. By contrast, a record of passing an officially recognized, sovereign-led test demonstrates that the provider met required safety standards before deployment. Instead of increasing exposure, the audit trail provides a formal defense by establishing a baseline of clinical responsibility. For providers seeking long-term stability in the medical market, this validation serves as a strong incentive for proactive participation.

# 8. Conclusion

Public medical benchmarks are no longer sufficient for certifying the safety of generative medical LLMs deployed in high-stakes clinical settings. The Open Benchmark Paradox ensures that as long as evaluation data remains public, performance scores will be undermined by hidden contamination and static clinical standards. To restore the scientific and legal validity of these evaluations, we must transition from global leaderboards to SME. By establishing assessment as a nationally governed safety infrastructure, SME replaces the disclosure of test data with a verifiable, evaluation-only access model. This shift provides the auditability necessary for legal accountability and ensures that AI behavior remains strictly aligned with local jurisdictional standards.

## Acknowledgements

This research was supported by a grant of the Korea Health Technology R&D Project through the Korea Health Industry Development Institute (KHIDI), funded by the Ministry of Health & Welfare, Republic of Korea (grant number: RS-2025-02213531).

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

# A. Detailed Settings for Data Contamination

## A.1. Dataset Preparation

The 2025 KMLE dataset was manually collected immediately following the official examination. To ensure a rigorous focus on text-based memorization and to maintain consistency across API-based models with varying multimodal capabilities, we excluded items containing images, diagrams, or charts. The final audit set consists of only text-based multiple-choice items. This filtering allows for a more precise measurement of n-gram overlap without the noise introduced by vision-to-language interpretation.

## A.2. Metric: Coverage Score

To quantify data contamination in a black-box setting, we utilize the Coverage Score as proposed by Hallinan et al. (2025), which is currently considered state-of-the-art for auditing API-based large language models. The coverage score measures the proportion of a reference text (the ground truth) that is reproduced by the model.

Given a model-generated completion $x_1$ and a reference suffix $x_2$, the coverage score $Cov_L(x_1, x_2)$ is calculated as follows:

$$Cov_L(x_1, x_2) = \frac{\sum_{w \in x_2} \mathbb{I}[\exists g \subseteq x_1 : w \in g \wedge |g| \geq L]}{|x_2|} \tag{1}$$

where $L$ represents the minimum n-gram length for a match. Following the recommendations of Hallinan et al. (2025), we set $L = 3$. To balance statistical reliability with computational cost, we extracted 30 independent completions per model for each test item across both experiments. The final scores reported are the averages of these 30 trials.

## A.3. Prompt Templates

Table 3 details the system and user prompts used for Experiment 1 (Choice Reconstruction) and Experiment 2 (Text Completion). The prompts are designed to minimize instructional bias while maximizing the model's reliance on its internal training data.

*Table 3.* Prompt Templates for KMLE Data Contamination Experiments.

| Exp. | Korean (Original) | English (Translated) |
|------|-------------------|----------------------|
| Exp 1 | **System:** 당신은 한국 의사 국가시험(KMLE) 형식에 익숙한 의학 전문가입니다.
**User:** 아래는 KMLE 스타일의 한 문제(stem)입니다. 이 문제에 어울리는 객관식 선택지 5개(A, B, C, D, E)를 작성하세요. | **System:** You are a medical expert familiar with the format of the Korean Medical Licensing Examination (KMLE).
**User:** Below is a KMLE-style question stem. Please write five multiple-choice options (A, B, C, D, E) that fit this question. |
| Exp 2 | **System:** 당신은 한국 의사 국가시험(KMLE) 스타일의 의학 문제 문장을 그대로 이어서 작성하는 전문가입니다.
**User:** 아래는 어떤 KMLE 문제 지문의 앞 50%입니다. 이 지문이 원래 존재했다고 가정하고, 남은 50% 뒷부분을 자연스럽게 이어서 작성하세요. | **System:** You are an expert at completing medical question texts in the style of the Korean Medical Licensing Examination (KMLE).
**User:** Below is the first 50% of a KMLE question stem. Assuming this text originally existed, complete the remaining 50% naturally. |

