# OpenReview forum: "Position: The Open Benchmark Paradox Must Be Resolved through Sovereign Medical Evaluation"
_ICML.cc/2026/Position_Paper_Track — ICML 2026 Position Paper Track regular_

### Official Review · Reviewer_cgqE · 2026-02-22

**Significance:** 3
**Argument Clarity:** 2
**Rating:** 4
**Confidence:** 4

**Questions:**

1. Currently, most countries have already established medical administration systems, although not specifically for AI. What is the relationship between SME and the existing systems?
2. What will be the basic unit of “sovereign”? For example, should the EU use a unified SME or one for each country? Should the US use a unified SME or one for each state?
3. What will be the costs of an SME system? Who should pay for it, and how much should be paid?
4. Is SME intended to complement rather than replace open benchmarks within a multi-stage evaluation ecosystem?

**Alternative Views Section:**

Yes

**Compliance With Llm Reviewing Policy A Conservative:**

Affirmed.

**Discussion Potential:**

3

**Final Justification:**

Thanks for the authors' response, which has addressed my concerns. I've raised my rating accordingly.

**Paper Summary:**

The paper argues that public medical benchmarks are inadequate for certifying clinical safety due to data contamination, outdated standards, and lack of jurisdictional alignment, and proposes Sovereign Medical Evaluation (SME), a regulator-controlled, isolated, and continuously updated national evaluation framework, as a more reliable safety gate for medical AI deployment.

**Position:**

Yes

**Position In Title:**

Yes

**Related Work:**

2

**Strengths And Weaknesses:**

**Strengths**:

1. This paper focuses on an essential challenge in medical evaluation.
2. The definition and illustration of the open benchmark paradox are clear.
3. The novel concept of sovereign medical evaluation is reasonable and potentially actionable.

**Weaknesses**:

1. For R2 (Governance), the evaluation process can also (and sometimes even more easily) be verified in public benchmarks, rather than only reporting the final score.
2. For R3 (Temporal Currency), many datasets and benchmarks are updated in a timely manner, such as PDB (Protein Data Bank) and MIMIC (Medical Information Mart for Intensive Care), which are maintained by established institutes and supported by many contributors. No sufficient justification is given to support that a sovereign system can be updated more efficiently than a well-maintained public dataset/benchmark.
3. The paper largely limits the discussion of medical AI to medical LLMs, while in practice AI for radiology and medical devices is more widely deployed. There are already regulatory frameworks for these systems that share similarities with the SME concept, such as the [FDA guidelines for AI-enabled medical devices](https://www.fda.gov/regulatory-information/search-fda-guidance-documents/artificial-intelligence-enabled-device-software-functions-lifecycle-management-and-marketing), so the contribution of SME may be somewhat overgeneralized.

Overall, the advocacy of SME is inspiring and promising, but at present, compared to existing sovereign or regulator-controlled evaluation systems in medicine and related fields, SME remains relatively high-level. It would be beneficial to learn from and explicitly compare with existing systems that are conceptually similar to SME.

**Support:**

2

---

> ### Author Rebuttal · Authors · 2026-03-28
>
> We deeply appreciate Reviewer cgqE for their critical and insightful assessment of the positioning and operational aspects of SME. We agree that the initial draft required a more precise situating of SME within the current regulatory landscape. In the revision, we will refine our claims to clarify SME’s role as a complementary framework.
>
> ---
>
> **1. Scope and relationship to existing regulatory systems (Weakness 3 & Question 1)**
>
> We agree that our initial draft inadvertently overgeneralized the scope of *medical AI.* In the revision, we will explicitly narrow our focus and terminology to generative *medical LLMs*.
> SME is not intended to replace existing regulatory authorities (e.g., FDA, MFDS); rather, it serves as a complementary technical evaluation framework integrated into existing pipelines. As the reviewer noted, current frameworks, such as the FDA guidelines for AI-enabled medical devices, are highly effective for traditional AI (e.g., radiology). These systems primarily function as evidence-support tools whose static outputs can be directly verified by clinicians.
> Generative medical LLMs, however, operate at a fundamentally different level of abstraction. They not only retrieve and summarize knowledge but also implicitly propose clinical reasoning and treatment plans. As a result, their outputs are not locally verifiable in the same way as image-based predictions, and clinicians may be cognitively influenced by the model’s reasoning rather than independently validating each step. This reflects a fundamental shift in the model’s epistemic role, from an evidence provider to a reasoning agent, which renders existing static validation frameworks insufficient.
>
> SME is specifically designed to address this gap by enabling dynamic, controlled evaluation of reasoning behavior under clinically relevant conditions. In this sense, SME complements existing validation regimes by targeting challenges unique to generative, language-mediated systems.
>
> ---
>
> **2. Governance: Public verification vs. SME (Weakness 1)**
>
> While public benchmarks excel in transparency and community-driven reproducibility, SME addresses a distinct objective: certification-grade auditability. As the reviewer noted, public benchmarks allow for pipeline verification, but the public nature of the data inevitably leads to contamination risks over time. SME is designed to preserve the chain-of-custody and independence of evaluation materials, ensuring that performance reflects genuine clinical competence. We will clarify this distinction in the manuscript to show that SME provides the rigorous evidence necessary for high-stakes certification.
>
> ---
>
> **3. Temporal currency: MIMIC/PDB vs. SME (Weakness 2)**
>
> We agree that well-maintained resources like MIMIC or PDB are highly efficient in their updates. Our claim regarding temporal currency, however, is focused on *update-to-certification integrity* rather than publication speed. Once new medical guidelines or safety alerts are released publicly, they risk being absorbed into the training corpora of future models. SME allows regulators to convert these updates into non-public, evaluative cases, maintaining the *unseen* status of the evaluation even for the most recent clinical developments.
>
> ---
>
> **4. Operational Clarifications (Q2, Q3, Q4)**
>
> We will incorporate the following operational details into Section 5 of the revised manuscript:
>
> - **The basic unit of sovereignty (Q2)**: We define *sovereign* functionally as any governance unit capable of enforcing standards of care and assigning medical liability, as detailed in Section 4. This could be national (e.g., South Korea), supranational (e.g., the EU), or subnational, depending on the legal structure of the jurisdiction.
> - **Costs and funding (Q3)**: We propose a user-fee model, similar to the FDA's user-fee programs for medical devices (e.g., MDUFA). Model providers seeking clinical deployment would contribute certification fees to sustain the SME infrastructure.
> - **Complementary role (Q4)**: We will clarify that SME is intended to complement, not replace, open benchmarks. Open benchmarks remain vital for general research and development, while SME provides the specialized layer required for rigorous clinical validation.
>
> By addressing these regulatory nuances and narrowing our focus to medical LLMs, we believe our revised manuscript offers a more actionable and grounded framework.

---

> > ### Author Rebuttal · Reviewer_cgqE · 2026-04-03
> >
> > Thanks for the author's response, which has addressed my concerns. I've raised my rating accordingly.

---

### Official Review · Reviewer_oFZm · 2026-03-05

**Significance:** 3
**Argument Clarity:** 3
**Rating:** 5
**Confidence:** 4

**Questions:**

My question above is about applicability of the proposed framework, and about providing examples since the paper had almost none. Examples would strengthen the paper a lot.

**Alternative Views Section:**

Yes

**Compliance With Llm Reviewing Policy A Conservative:**

Affirmed.

**Discussion Potential:**

2

**Final Justification:**

I'll keep my accept score.

**Paper Summary:**

This paper's position is that sovereign evaluations can certify LLMs for medical usage. The paper discusses why open benchmarks are problematic, because the LLM has probably seen the evaluation already, and because things change and it needs to adapt, and because different locations require different models. It goes through quite a lot of recommendations, including jurisdiction-specific benchmarking, signal isolation to prevent private data leakage, different testing procedures, and more.

**Position:**

Yes

**Position In Title:**

Yes

**Related Work:**

3

**Strengths And Weaknesses:**

Strengths: I thought this paper had some interesting points in it and gave quite a lot of recommendations on how to evaluate medical systems. It also gave useful alternative views.

Weaknesses:
One thing I'd like the authors to consider is the usage of the LLMs and whether the full proposed process needs to apply to all LLMs. If every LLM for medical data usage needs to go through this entire process, it would prevent specialized LLMs for being used, even if there is a narrow low-stakes scope where the LLM could reliably do well on a specific task (and it doesn't matter if it doesn't do well because it's low-stakes). If you impose blanket restrictions, it really would stop a lot of medical research. I think clarifying the scope of your desired restrictions would help a lot - give examples of LLM applications that would (and would not) need to satisfy your testing criteria. There really are no examples in this paper except for the test questions, which makes one wonder what applications these test are needed for. The paper does get quite dull to read in the middle with all these evaluation lists, so perhaps some examples would make it better to read.

**Support:**

3

---

> ### Author Rebuttal · Authors · 2026-03-28
>
> We sincerely thank Reviewer oFZm for the helpful feedback regarding the scope and readability of our proposal. We agree that without a clearly defined scope, SME could be misinterpreted as a blanket restriction that might inadvertently constrain innovation. In the revision, we will introduce a risk-tiered framework and clinical vignettes to clarify the practical application of SME.
>
> ---
>
> **1. Risk-Tiered Applicability Framework**
> We will clarify that SME is not intended as a universal requirement for all medical LLM applications. To avoid unnecessary constraints on academic research and preserve academic freedom, we have established a risk-tiered framework in the revised manuscript:
> - **High-stakes clinical deployment (SME required)**: This applies strictly to models seeking certification for direct clinical actions where errors involve legal liability and patient safety risks. Examples include autonomous triage, real-time dosing suggestions, and automated diagnostic assistants.
> - **Low-stakes research and education (SME exempt)**: Applications with a narrow scope or specialized LLMs utilized for low-stakes tasks will continue to operate under existing institutional reviews and open benchmarks. Examples include medical literature summarization, information extraction from clinical notes, and non-deployment R&D prototypes.
>
> By explicitly separating these tiers, we ensure that SME targets only the liability-prone areas of clinical practice without hindering broader medical research or the development of specialized, narrow models.
>
> ---
>
> **2.  Enhancing readability with concrete clinical vignettes**
> The reviewer’s feedback regarding the readability of the evaluation lists is well-taken. To clarify the necessity of SME and make the manuscript more engaging, we will supplement the abstract descriptions with concrete clinical vignettes. These examples demonstrate scenarios where open benchmarks may be insufficient, and SME becomes essential. Below are two examples added to the revision:
>
> *Example Vignette 1 (Dynamic Regulatory Adherence)*
> - Consider a scenario where a health authority issues a new 'black-box warning' for a cardiovascular drug due to newly discovered risks. A model relying on static pre-training data or open benchmarks might continue to recommend the drug. Under the SME framework, this regulatory update is immediately translated into a private clinical case (e.g., 'A 65-year-old patient with [newly identified contraindication] presents with... What is the best next step?'). Crucially, SME evaluates critical clinical reasoning rather than blind compliance. An ideal model response would not merely withhold the drug but acknowledge its traditional role as a standard of care, explain the newly identified risk specific to the patient, and proactively recommend a safer second-line alternative. This tests whether an LLM can dynamically override its outdated parametric knowledge and synthesize the latest safety mandates into a nuanced care plan, a task that static benchmarks cannot perform.
>
> *Example Vignette 2 (Jurisdictional Protocol Adherence)*
> - To test applications designed for automated treatment recommendations, another vignette introduces a patient requiring a specific therapy that is standard in one country but unapproved in the target jurisdiction (e.g., Radotinib in Korea vs. Western protocols, as discussed in Section 2.3). This ensures the LLM adheres to local medical guidelines rather than generic global consensus.
>
> We believe these additions provide a more nuanced and rigorous policy framework. We appreciate the reviewer's guidance in making our contribution more actionable and readable.

---

> > ### Author Rebuttal · Reviewer_oFZm · 2026-04-01
> >
> > I appreciate the responses - I think making those changes would improve the paper. I'll keep my positive score.

---

### Official Review · Reviewer_MURp · 2026-03-10

**Significance:** 3
**Argument Clarity:** 3
**Rating:** 4
**Confidence:** 3

**Questions:**

See weakness

**Alternative Views Section:**

Yes

**Compliance With Llm Reviewing Policy A Conservative:**

Affirmed.

**Discussion Potential:**

3

**Final Justification:**

My concern has been resolved, I keep my positive score

**Paper Summary:**

This position paper argues that public medical benchmarks have become structurally inadequate for certifying clinical safety due to the Open Benchmark Paradox, where the public availability of evaluation data makes data contamination inevitable and ruins its value as a reliable safety signal. Through an audit of frontier models like GPT-5 and Gemini 3 Pro using the 2025 Korean Medical Licensing Examination, the authors provide empirical evidence of "hidden contamination," showing that these models possess near-verbatim knowledge of recent test items likely ingested during automated training data collection. To resolve this, they propose Sovereign Medical Evaluation (SME), a national-level infrastructure that replaces public leaderboards with private, jurisdictionally governed evaluation pipelines. SME ensures safety by mandating strict data isolation through evaluation-only API access, using automated pipelines to update test scenarios from live national health sources, and anchoring assessments in the specific legal and ethical mandates of a local jurisdiction.

**Position:**

Yes

**Position In Title:**

Yes

**Related Work:**

3

**Strengths And Weaknesses:**

Strength:

1. The authors provide strong quantitative evidence by auditing frontier models (GPT-5, GPT-5.1, Claude 4.5 Sonnet, and Gemini 3 Pro) using the 2025 Korean Medical Licensing Examination.

2. It correctly identifies that medical safety is not a universal metric but is deeply tied to local legal mandates, drug approvals (e.g., radotinib in Korea vs. Western standards), and ethical norms.

3. The authors provide a discussion of credible alternative views, such as concerns over scientific fragmentation and global health equity, which strengthens the paper's position.

Weakness

1.  The proposed Sovereign Medical Evaluation (SME) requires a national-level infrastructure that may be technically and administratively difficult for academics to be involved in.

2. One thing to avoid contamination is time cliff data, where the data is after the model training time. So, in addition to the metric,  maintaining a rolling evaluation dataset will complete the SME, so it needs some discussion.

**Support:**

3

---

> ### Author Rebuttal · Authors · 2026-03-28
>
> We are grateful to Reviewer MURp for their insightful and constructive comments. We particularly value the suggestions regarding *time-cliff data* and *rolling evaluation*, which provide a clear operational path for SME. We will incorporate these concepts into the revised manuscript to enhance the practical viability of our framework.
>
> ---
>
> **1. The Role of Academics in SME (Weakness 1)**
>
> We acknowledge the concern that national-level infrastructure might appear to limit the involvement of the academic community. We will clarify that SME is envisioned as a partnership model where academics serve as primary architects rather than acting solely as administrative operators. In the revision, we will detail how this collaboration functions:
>
> - **Academics as primary architects**: While state authorities provide the jurisdictional mandate and secure hosting, the academic community is responsible for designing the scientific core. This includes developing automated red-teaming pipelines, dynamic scenario generation, and contamination-detection heuristics.
> - **Multi-stage ecosystem**: SME is not intended to replace open science. Open benchmarks remain the primary venue for foundational academic R&D. SME serves as a specialized regulatory gateway for models seeking formal clinical certification. This structure ensures that academic inquiry and early-stage innovation are preserved while maintaining rigorous standards for clinical deployment.
>
> ---
>
> **2. Incorporating Time-Cliff Data and Rolling Evaluation (Weakness 2)**
>
> Following the reviewer's valuable suggestion, we will incorporate the concepts of *time-cliff data* and *rolling evaluation* to provide a principled approach to mitigating contamination. To better capture the clinical characteristics of the medical domain, we propose a three-tiered stratified rolling protocol, designed to remain robust under the power-law distribution of clinical cases.
> Rather than relying on a uniform rolling mechanism, our proposed protocol (detailed in Section 4.2 of the revision) organizes evaluation data into three complementary strata:
> - **Tier 1 (Temporal stratum)**: For newly approved drugs and updated clinical guidelines, we adopt a strict time-cliff protocol. This ensures that models are evaluated on information that post-dates their training sets.
> - **Tier 2 (Core stratum)**: For high-prevalence diseases, we maintain the rolling property by procedurally varying clinical vignettes (e.g., demographics, comorbidities) rather than altering the underlying diagnosis. This enhances evaluation by preventing reliance on memorized templates.
> - **Tier 3 (Adversarial stratum)**: We continuously incorporate atypical presentations and rare contraindications to evaluate safety limits rather than pattern recall.
>
> By integrating these mechanisms, SME establishes a sustained evaluation framework via rolling datasets, creating a continuous mitigation cycle that systematically bounds the risk of data contamination and memorization.

---

> > ### Author Rebuttal · Reviewer_MURp · 2026-03-31
> >
> > My concern has been resolved, I keep my positive score

---

### Official Review · Reviewer_bB4o · 2026-03-11

**Significance:** 2
**Argument Clarity:** 2
**Rating:** 4
**Confidence:** 4

**Questions:**

How to guarantee the capacity of the sovereign infrastructure? Participants may blame and attribute the inferior performance to the data host, e.g., cannot run their very large 500B model, run their model properly (given a complex pipeline), or use a correct random seed.

**Alternative Views Section:**

Yes

**Compliance With Llm Reviewing Policy A Conservative:**

Affirmed.

**Discussion Potential:**

2

**Final Justification:**

Most concerns have been responded to with rationale, though the second point might require substantial revision.

**Paper Summary:**

This paper discusses the position of a false sense of LLM performance on open benchmark, largely due to the issue that data have already been seen by LLM in some steps of model training, e.g., data contamination.

The paper advocates a shift from global leaderboards toward Sovereign Medical Evaluation: a national safety infrastructure that is jurisdictionally governed, auditable, and continuously updated.

**Position:**

Yes

**Position In Title:**

Yes

**Related Work:**

3

**Strengths And Weaknesses:**

Strength:

+ Open benchmark paradox is an important challenge in ML community.
+ The paper discusses three structural failures under the paradox.
+ The paper suggests a rational solution to this by hosting infrastructure where health authorities manage private and isolated evaluation pipelines.


Weaknesses:

- The main question is that the paper discusses the paradox with a focus on a narrow (medical) domain, while I believe it is a generic problem in arguably almost all areas applying LLM. The paper should clearly position the question with similar questions in other relevant domains, like legal, education, finance, industry, etc. What is the difference and corresponding challenge in the medical domain?

- The solution is interesting, though model providers would also consider their models as valuable assets and would be unwilling to provide their models to evaluators. Introducing "mid-party" or "third-party" would be considered a more interesting solution from my point of view. Providing such a discussion would significantly improve this work.

- It is not easy to guarantee and differentiate whether the data is fully unseen by the model. Even if the data are hosted privately, some sections could arguably be observed or even used for training as well.

**Support:**

3

---

> ### Author Rebuttal · Authors · 2026-03-28
>
> We truly appreciate Reviewer bB4o’s thoughtful and constructive feedback. We especially value the suggestion regarding a *mid-party* framework, which we agree is essential for the practical implementation of SME.
>
> ---
>
> **1. Medical Domain Specificity (Weakness 1)**
>
> While we acknowledge that data contamination is a broad issue across domains such as finance and law, we argue that the medical domain presents a unique necessity for SME due to the direct link between model output and physical safety. Unlike errors in other sectors that primarily result in financial or systemic friction, medical inaccuracies involving locally unapproved treatments, such as Radotinib's status in Korea versus Western standards (Section 2.3), can lead to fatal consequences. This necessitates jurisdiction-specific governance that extends beyond general open benchmarks. We will clarify this positioning in the revision to highlight that while the challenge of contamination is pervasive, the stakes and regulatory requirements in medicine are uniquely rigorous.
>
> ---
>
> **2. Mid-party Framework and Infrastructure (Weakness 2 & Question 1)**
>
> Following the reviewer's valuable suggestion, we will incorporate a *mid-party* framework into our proposal. This approach effectively addresses the tension between protecting model IP and ensuring robust jurisdictional oversight. In the revised manuscript, we will clarify that SME facilitates governance over the evaluation process rather than requiring state ownership of compute or model weights. This can be operationalized through accredited third-party evaluators or provider-hosted confidential endpoints under jurisdictional control.
>
> To address the reviewer’s inquiry regarding infrastructure capacity and performance disputes, we will introduce an SME operational protocol based on this framework in the revision:
> - **Pre-registered evaluation manifest**: Providers formally declare model parameters (e.g., version hash, tokenizer, and hardware tier) before evaluation.
> - **Dry-run validation**: Providers conduct preliminary execution on non-secret calibration tasks to verify pipeline compatibility.
> - **Attested execution logs**: While test items remain private, all execution states and resource usage are logged for auditability.
>
> By integrating this framework, the computational burden remains with the provider. Therefore, providers can seamlessly run their very large models (e.g., 500B parameters) on their own optimized infrastructure, mitigating capacity constraints on the evaluator’s side while ensuring the evaluation conditions are transparent and legally auditable.
>
> ---
>
> **3. Guaranteeing Unseen Data (Weakness 3)**
>
> While we acknowledge that ensuring a complete absence of data contamination is a pervasive challenge in the field, we will explicitly discuss this limitation in the revision. Rather than claiming an absolute guarantee, we frame SME as a method to mitigate exposure and provide a more rigorous institutional standard of diligence. Under the mid-party API setup, evaluators can enforce strict zero-logging agreements or utilize confidential computing (e.g., secure enclaves) to prevent providers from directly observing or extracting the private test queries for training. By utilizing these mechanisms alongside non-public updates, SME significantly reduces the risk of intentional or accidental data harvesting compared to open benchmarks, offering a more robust framework for high-stakes deployment.

---

> > ### Author Rebuttal · Reviewer_bB4o · 2026-04-03
> >
> > Most concerns have been responded to with rationale, though the second point might require substantial revision. Shepherd acceptance might be more suitable for this case; however, ICML doesn't provide such an option. Considering the situation, I decided to upgrade my overall score to 4.

---

### Decision · Program_Chairs · 2026-04-30

**Decision:**

Accept (regular)

**Comment:**

While reviewers agree that the underlying problem of the Open Benchmark Paradox is a significant challenge for the ML community, they also raise substantive concerns regarding the scope of the paper that may limit interest. In particular, the paradox itself is a broad challenge for the community but the position hold a relatively narrow focus on the medical domain. While this is acknowledged in rebuttal, the underlying concern is that the narrowed scope will no longer be engaging to the broader general domain audience of ICML. A possible modification to address this might be a presentation of the larger issue with medicine framed as a salient, illustrative case study. However, as currently framed the audience of interest is comparative limited to the clinical subcommunity, which may be already aware of these challenges and unconvinced of the position due to its dependency on national-level infrastructure that is typically developed by other communities. An additional concern is that existing regulatory evaluation might hold parallels that should be discussed. This is partially addressed in rebuttal, though there would ideally be additional detail in the response about how this could be incoroporated. Taken together, these concerns limit the potential interest of the underlying work, though I would certainly encourage the authors to incorporate the reviewer's feedback in future revision.